# Thermal Behavior of Pea and Egg White Protein Mixtures [note 1]

**DOI:** 10.3390/foods12132528

**Published:** 2023-06-29

**Authors:** Jian Kuang, Pascaline Hamon, Valérie Lechevalier, Rémi Saurel

**Affiliations:** 1PAM UMR A 02.102, L’Institut Agro Dijon, Université Bourgogne Franche-Comté, F-21000 Dijon, France; jian_kuang95@yeah.net; 2INRAE, L’Institut Agro Rennes-Angers, UMR STLO, F-35042 Rennes, France; pascaline.hamon@agrocampus-ouest.fr (P.H.); valerie.lechevalier@agrocampus-ouest.fr (V.L.)

**Keywords:** pea protein isolate, egg white protein, differential scanning calorimetry, gelling point, solubility, coagulation

## Abstract

The partial substitution of animal protein by plant protein is a new opportunity to produce sustainable food. Hence, to control the heat treatment of a composite protein ingredient, this work investigated the thermal behavior of mixtures of raw egg white (EW) and a laboratory-prepared pea protein isolate (PPI). Ten-percentage-by-weight protein suspensions prepared with different PPI/EW weight ratios (100/0, 75/25, 50/50, 25/75, 0/100) at pH 7.5 and 9.0 were analyzed by differential scanning calorimetry (DSC) and dynamic rheology in temperature sweep mode (T < 100 °C). The DSC data revealed changes in the thermal denaturation temperatures (Td) of ovotransferrin, lysozyme, and pea legumin, supposing interactions between proteins. Denaturation enthalpy (∆H) showed a high pH dependence related to pea protein unfolding in alkaline conditions and solubility loss of some proteins in admixture. Upon temperature sweeps (25–95 °C), the elastic modulus (G′) of the mixtures increased significantly with the EW content, indicating that the gel formation was governed by the EW protein. Two thermal sol–gel transitions were found in EW-containing systems. In particular, the first sol–gel transition shifted by approximately +2–3 °C at pH 9.0, probably by a steric hindering effect due to the presence of denatured and non-associated pea globulins at this pH.

## 1. Introduction

Owing to the population growth and diet-related socioeconomic changes over the coming decades, humans are increasingly recognizing that a greater consumption of plant-based foods and less dependence on meat and other animal-based products will contribute to improving the sustainability of the food system [1,2,3,4,5,6,7]. Meanwhile, the food industry is increasingly using plant protein components, particularly from legume seeds, as an alternative to animal-based sources due to their diversity, nitrogen-fixing ability, higher availability, low price, and consumer perception of health and sustainability [8,9]. The mixtures of plant and animal proteins have also been considered to address food transition concerns and to explore synergistic effects in terms of consumer acceptance, nutrition, digestibility, and techno-functional properties of such systems [10,11]. Among the new sources of proteins, pea proteins are increasingly attractive, and several studies have targeted their behavior when mixed with dairy proteins to form gels, emulsions, or foams [12,13]. On the other hand, no study has explored mixtures of pea and egg proteins, which may be an interesting perspective for the development of mixed protein ingredients and derived ovo-vegetarian products.

Pea proteins represent ≈23% of dry seeds [14,15] and are mainly composed of globulins (≈70%), and the rest corresponds mainly to the 2S albumin fraction (≈20%) and other minor insoluble proteins [16]. The globulin fraction containing mainly legumin 11S and vicilin/convicilin 7S can be recovered by alkaline extraction and isoelectric precipitation at pH 4.5–4.8 to produce pea protein isolate [17]. Legumin is a hexameric protein of 360–400 kDa, comprising six subunits of ~60 kDa associated with non-covalent interactions. Each legumin monomer consists of an acidic (~40 kDa) and an alkaline (~20 kDa) subunit linked by a disulfide bond [18,19]. Vicilin (7S) is a trimeric protein of around 150 kDa. Each monomer ~50 kDa has two cleavage sites possibly generating small fragments during pea seed development: α (~20 kDa), β (~13 kDa), γ (~12–16 kDa), αβ, and βγ polypeptides [15,19,20,21]. A third minor globulin, convicilin, is a multimeric protein of 210–290 kDa whose subunit (~70 kDa) has a highly homologous core amino acid sequence with the vicilin monomer, yet possesses an extended hydrophilic N terminus [22].

Among animal-rich protein products, egg white is a desirable ingredient used in many foods, such as bakery products, meringues, and meat products, because of its excellent foaming and gelling properties [23]. It usually contains approximately 11% proteins, which consist of more than 40 different kinds of proteins. Ovalbumin (54%), ovotransferrin (12%), ovomucoid (11%), lysozyme (3.5%), and ovomucin (1.5–3.5%) are among the major proteins of egg white [24,25]. Ovalbumin (44.5 kDa) consists of a peptide chain of 385 amino acid residues and contains 4 thiols and 1 disulfide group. Its isoelectric point (pI) is estimated at 4.5. With a high pH and temperature-dependent denaturation, ovalbumin converts into a thermally stable form known as S-ovalbumin [26]. Ovotransferrin consists of 1 peptide chain of 686 amino acids and contains 1 oligosaccharide unit made of 4 mannose and 8 N-acetylglucosamine residues. Its molecular weight and pI are around 77.7 kDa and 6.1, respectively. Lysozyme (14.4 kDa) is a relatively small secretory glycoprotein consisting of 129 amino acids linked by 4 disulfide bonds with an isoelectric point of 10.7. Recently, Iwashita [27] highlighted that ovotransferrin can co-aggregate with lysozyme, and Wei [28] confirmed heteroprotein complex formation between ovotransferrin and lysozyme. Ovomucoid is a heat-stable glycoprotein containing 186 residues with a molecular weight of 28 kDa and pI of 4.1 [29,30]. It contains nine disulfide bonds and has three different domains that are crosslinked only by the intra-domain disulfide bonds [31]. Ovomucin is a sulfated glycoprotein that is responsible for the jelly-like structure of egg white [32,33]. The molecular weight of ovomucin is 1.8–8.3 × 103 kDa [34,35]. Based on their composition and characteristics, ovomucin subunits may be classified into two types: α- and β-ovomucin [31].

At an industrial level, egg white is available in pasteurized liquid or frozen form, or powder after spray drying. During processing, the egg white is pasteurized at moderate temperatures between 54 and 57 °C for a few minutes to prevent the coagulation of the heat-sensitive egg white proteins. The egg white can also be subjected to higher temperatures during spray drying operating up to 180 °C, and the dry heating of egg white powder up to 80 °C is commonly applied to improve its microbiological quality and technofunctional properties [23]. From the perspective of developing a mixed ingredient combining egg white and pea proteins and undergoing treatments similar to those applied to commercial egg products, it seems necessary to evaluate the thermal behavior of the mixture from native proteins to prevent inappropriate protein denaturation or gelation. To our knowledge, egg proteins have rarely been studied in association with plant proteins except with soybean protein [36,37], and the physicochemical properties of egg white and pea protein mixture have not yet been considered in previous works.

Therefore, this paper aimed to evaluate the thermal properties and heat-induced denaturation of pea protein isolate (PPI) and egg white (EW) mixtures at different weight ratios at alkaline pH by differential scanning calorimetry. Meanwhile, the nitrogen solubility profile of the pure proteins and 50/50 mixture systems and their protein composition using electrophoresis, as well as the gelling temperature of different 10 wt% PPI-EW mixtures studied by small amplitude rheology at pH 7.5 and 9.0, were also investigated to elaborate the thermal behavior of the composite protein systems. This study may not only provide a basic knowledge of the hybrid PPI and EW system but also anticipate a possible adjustment in treatment temperature in the future manufacturing of such a mixed ingredient.

## 2. Materials and Methods

### 2.1. Materials

Pea globulins were extracted from smooth yellow pea flour (*P. sativum* L.) supplied by Cosucra (Warcoing, Belgium). Fresh eggs (Label Rouge, Dijon, France) were obtained from a local market (Dijon, France), stored in a fridge at 4 °C, and used 15 days before the expiration date. All other reagents and chemicals purchased from Sigma-Aldrich (St-Quentin Fallavier, France) were of analytical grade.

### 2.2. Pea Protein Extraction

The isoelectric precipitation technique was used to prepare pea protein isolate (PPI), containing mainly globulins, based on the method of Chihi [12] with some modifications. Pea flour was mixed with distilled water at 100 g/L, and the pH was adjusted to pH 8.0 with concentrated NaOH (≤0.5 M) every 2 h (3 times) and stirred overnight at 4 °C. After adjusting the final pH to 8.0, insoluble materials were removed by centrifugation (10,000× *g*, 30 min, 20 °C) using a SORVALL R6 PLUS centrifuge (Thermo Electron Corporation, Saint Herblain, France) and the recovered solution was adjusted to pH 4.8 by 0.5 M HCl. After acidification, the precipitated proteins were separated by centrifugation (10,000× *g*, 25 min, 4 °C). Afterward, the pellets were dissolved in 5 L of 0.1 M phosphate buffer at pH 8.0 overnight at 4 °C for complete dissolution. The protein suspension was obtained by centrifugation (10,000× *g*, 20 min, 20 °C) and then concentrated 5 times by ultrafiltration (from 5 L to 800–900 mL) at room temperature using a Pellicon^®^ 2 Mini-Holder (Millipore, Dachrstein, France) equipped with an ultrafiltration 1115 cm^2^ Kvick lab Cassette (UFELA0010010ST, GE Healthcare, Amersham Biosciences, Uppsala, Sweden) with a molecular weight cut-off of 10 kDa. The resulting concentrate was consecutively desalted by diafiltration against 10 L of 5 mM ammonium buffer pH 7.2 and 0.05% sodium azide on the same device. Protein powder as PPI was obtained by freeze-drying as follows. The diafiltered protein suspension was frozen overnight at −80 °C in sampling vessels filled with 1 to 1.5 cm thickness liquid. The freeze-drying step was then performed for 72 h at −57 °C at a pressure lower than 1 hPa using a Heto Power Dry PL 6000 freeze-dryer (Thermo Electron Corporation, Waltham, MA, USA). The protein content of PPI was measured at 89% on a dry basis by the Kjeldahl method (N = 5.44).

### 2.3. Sample Preparation

Suspensions of PPI were prepared by dissolving the freeze-dried powder in distilled water at a protein concentration of 10 wt% close to the protein concentration of EW protein. The dispersion was then agitated at 350–400 rpm for more than 3 h at 4 °C to allow for the complete hydration of proteins. Eggs were broken and the fresh liquid EW was carefully separated manually from egg yolks and chalaza. The obtained egg white was then put in a beaker and gently homogenized by a magnetic stirrer for 2 h at room temperature. The total protein content of egg white was determined by the Kjeldahl method (N = 6.25, 10.2%). The suspensions of PPI-EW mixture (protein concentration, 10 wt%) were then prepared by mixing PPI suspension and EW at various weight ratios of 100/0, 70/30, 50/50, 30/70, and 0/100. Then, the pH of the samples was adjusted to 7.5 or 9.0 using 1 M NaOH or 1 M HCl. Both alkaline pHs allowed for high soluble protein content and covered the range of EW pH during egg storage.

### 2.4. Solubility

The protein solubility of PPI, EW, and the PPI-EW mixture at a weight ratio of 50/50 was determined as a function of pH by the method described by Djoullah [38]. Protein suspensions (1%, *w*/*v*) were prepared in distilled water. The pH of the suspensions was adjusted from pH 2 to 10 with either HCl or NaOH (0.1 M). In addition, 0.1 M NaCl was used to maintain the ionic strength in EW diluted condition. After stirring for 2 h at 4 °C, mixtures were centrifuged (10,000× *g*, 20 min), and the nitrogen content of the supernatant was measured by the Kjeldahl method. The protein solubility as nitrogen solubility was determined by using Equation (1):NS(%) = (N dissolved in the supernatant)/(N initially in the suspension) × 100%(1)
where NS is nitrogen solubility and N is nitrogen amount.

Meanwhile, the expected solubility at each pH was used as a reference for the solubility of PPI-EW system (at a weight ratio of 50/50) and calculated by Equation (2):solubility_expected_ = solubility_EW_ × 0.5 + solubility_PPI_ × 0.5(2)

### 2.5. Sodium Dodecyl Sulfate–Polyacrylamide Gel Electrophoresis (SDS-PAGE)

The protein composition of PPI, EW, and PPI-EW samples at a weight ratio of 50/50 at pH 7.5 and 9.0 was characterized by SDS-PAGE. Novex™ electrophoresis gels at 10% to 20% Tris–glycine were used. Samples were diluted by at least half in sample buffer: 187.5 mM Tris-HCl, pH 8.9, 10% (*w*/*v*) glycerol, 2% (*w*/*v*) SDS, and 0.05% (*w*/*v*) bromophenol blue, in the presence (reducing condition) or absence (non-reducing condition) of 2% (*w*/*v*) dithiothreitol (DTT). The samples under reducing conditions were heated in a water bath for 10 min at 95 °C. All the samples were prepared and then 10 µg of protein was deposited. Molecular weight protein markers from Sigma–Aldrich^®^ (SigmaMarker™ S8445, wide range, Mw 6.5 to 200 kDa) or Thermo Scientific™ (PageRuler™ Unstained Broad Range Protein Ladder, Mw 5 to 250 kDa) were used. The migration was carried out at 35 mA per gel, with the following migration buffer: 0.3% (*w*/*v*) trizma base, 1.45% (*w*/*v*) glycine, and 0.1% (*w*/*v*) SDS, in a Scientific^®^ Mini Gel Tank of Migration (Thermo Fischer Scientific, Waltham, MA, USA). The gels were then rinsed with distilled water, and the fixation was performed in 4 successive distilled water baths heated for 1 min in a microwave at 550 W. The staining of the gels was performed with Coomassie blue (Thermo Scientific™ PageBlue™ Protein Staining Solution) overnight. The discoloring was then achieved in several baths of distilled water until the desired color. The gels were imaged by the ChemiDoc™ XRS + System from Bio-Rad Lab (Marnes-la-Coquette, France). To know the difference in protein composition of the PPI-EW mixture with or without centrifugation, a centrifugation step (10,000× *g*, 20 min, 4 °C) was alternatively performed on protein suspensions before analysis.

### 2.6. Differential Scanning Calorimetry (DSC)

Protein denaturation was assessed by DSC. The temperature of denaturation (Td) and enthalpy of denaturation (ΔH) were determined using a Micro DSC III calorimeter (Setaram, Caluire, France). PPI-EW mixture suspensions at 10 wt% protein were analyzed at different weight ratios (0/100, 25/75, 50/50, 75/25, 100/0). Approximately 0.5 g of sample was weighed in an aluminum pan, hermetically sealed, and heated from 25 to 105 °C at 0.5 °C/min. A pan containing distilled water was used as a reference. All experiments were conducted in triplicate. One replicate of each sample was re-heated after cooling to check that denaturation was irreversible. Deconvolution of thermograms performed by Origin 2019 Pro and thermal analysis SETSOFT 2000 V3.0 software (Setaram, Caluire, France) was used to identify the peaks and determine the thermal parameters more precisely.

### 2.7. Small-Strain Dynamic Rheology

Dynamic rheology was applied to evaluate the gelation characteristics of the protein systems. Ten-percentage-by-weight protein suspensions of PPI, EW, and PPI-EW at different weight ratios (75/25, 50/50, and 25/75) were prepared at pH 7.5 and 9.0. Subsequently, each sample was loaded into a rheometer MCR 302e (Anton Paar, Graz, Austria) equipped with a plate–plate geometry (50 mm diameter). Approximately 1 mL of the protein suspension was transferred to the lower plate of the rheometer. The upper plate was lowered to give a gap width of 1 mm. A thin layer of light mineral oil was added to the well of the upper plate geometry and a solvent trap cover was used to prevent sample drying during heating. The following heating protocol was used. Samples were first equilibrated at 25 °C for 3 min, then heated under 1% shear strain and 1 Hz of frequency over a temperature range of 25–95 °C at a rate of 2 °C/min (heating ramp). Rheological data as storage modulus (G′) and loss modulus (G″) were collected for every degree change during heating. The thermal gelation profile of EW, i.e., G′ as a function of temperature, obtained at pH 9.0 was chosen as a typical curve to illustrate how the data were analyzed (Figure 1). In this case, we observed two inflection points at ~58 and ~75 °C, respectively, where the G′ values rose sharply and diverged significantly from the loss modulus values (G′ >> G″). The points on both sides of one inflection point were fitted by linear models and the intersection of the respective straight lines was considered as the gelling temperature or gelling point as represented in Figure 1. Runs and analyses were performed in triplicate for each sample.

### 2.8. Statistical Analysis

Differences between samples were studied by analysis of variance (one-way ANOVA). Significance was set at *p* < 0.05. Tukey’s post hoc least-significant-differences method was used to describe means with 95% confidence intervals. The statistical analyses were performed using Statistica software, version 12 (Tulsa, OK, USA).

## 3. Results

### 3.1. Solubility Profile of Protein Systems

The solubility of PPI, EW, and the PPI-EW mixture at the 50/50 weight ratio is shown in Figure 2. Firstly, the solubility of PPI showed a U-shaped profile with pH. Similar pH-dependent protein solubility profiles of pea protein have been observed for commercial or native PPI [19,21,39,40,41] or pea protein concentrate [8]. In detail, PPI solubility passed through a minimum (~7%) at around pH 5, which is close to the isoelectric point (pI~4.5–5.0) of pea globulins, in agreement with the previous works. Shevkani [42] reported that the solubility of the tested pea protein isolate was only 2–4% at pH 5.0. Outside this range, the solubility increased and reached values of around 85% and 89% for pH values below 3.0 and above 7.0, respectively. This solubility behavior of globulins is attributed to electrostatic repulsion and the hydration of charged residues [43]. Around pI, the protein has as many positive charges as negative charges, which promotes the affinity and interaction between proteins at the expense of protein–solvent, thus minimizing solubility. For a more acidic and alkaline pH, the protein is positively and negatively charged, respectively. This large net charge increases the repulsion forces, thereby promoting the protein–solvent interaction, which results in better solubility. For egg white, the solubility was always higher than 88%. However, a little lower solubility of EW was observed around pH 4.0, which is close to the isoelectric point of ovalbumin (pH 4.5), the major egg white protein. This result is in good agreement with the previously reported solubility profiles of egg white and attests to the high hydrophilicity of egg white proteins [44,45]. Regarding the PPI-EW mixture, the solubility profile showed the same shape compared to PPI, with a minimum solubility (~55%) around pH 5.0.

Meanwhile, calculated NS values for the mixture based on the data of pure protein systems gave a reference (Table 1). When the pH was smaller than 4.0, the measured NS values of the 50/50 mixture showed no significant difference from the calculated ones. However, at pH 5.0, the measured NS value was significantly higher than the calculated one, while, above this pH, the NS profile showed significantly lower real values. The mixture of both proteins could either favor protein solubility (around pH 5.0) by a salting-like effect or enhance the formation of insoluble (co-)aggregates that decreased the protein solubility above pH 5.0. For instance, it has been proven that, at pH > 7, pea globulins form aggregated complexes with lysozyme through ionic interactions that can affect pea globulin solubility [46]. For the following steps of the study, pH 7.5 and 9.0 were retained as alkaline pHs, corresponding to a high protein solubility of the mixed system and covering the range of pHs of fresh or processed EW [23].

### 3.2. Protein Composition

Figure 3 shows the electrophoresis profile of PPI and EW under reducing and non-reducing conditions at pH 7.5 and 9.0. Figure 4 shows the electrophoresis profile of the 50/50 weight ratio PPI-EW mixture before and after centrifugation under reducing and non-reducing conditions at pH 7.5 and 9.0. In general, the electrophoretic profile of the PPI and EW prepared at pH 7.5 and 9.0 showed great similarities (Figure 3). Regarding PPI (non-reducing conditions, lanes 1 and 5, reducing conditions, lanes 2 and 6, at pH 7.5 and 9.0, respectively), the band observed at ~88 kDa was probably lipoxygenase [47,48]. A marked band at around 60 kDa corresponded to the legumin 11S main subunits (Lαβ), which were dissociated into acid Lα (~38–40 kDa) and basic Lβ (~20–22 kDa) subunits under reducing conditions as also observed elsewhere [19,21,49,50]. Different bands within the 15–50 kDa range present under reducing and non-reducing conditions were considered to include polypeptides of vicilin: (i) a large band close to 50 kDa could be assigned to the vicilin monomer [47]; (ii) 4 other polypeptide bands in the range of 15–37 kDa were supposed to correspond to fragments γ (12–16 kDa), α (~18 kDa), β:γ (~25–30 kDa), and α:β (~30–36 kDa) [20,47,51]. The convicilin monomer was present at ~70 kDa as expected [21,52]. Non-migrating aggregates (Mw > 250 kDa) were found (Figure 3, lanes 1 and 5) and they disappeared in reducing conditions. These disulfide bonded aggregates were probably produced during the extraction process as proposed in the study by Karaca [53].

Three main protein components were shown in egg white samples (Figure 3, lanes 3 and 7 at pH 7.5 and 9.0, respectively): ovotransferrin (76 kDa), ovalbumin (44 kDa), and lysozyme (14.6 kDa), in agreement with previous works [54,55]. No band corresponding to ovomucoid (∼28 kDa) could be visualized on the gel. As expected, ovalbumin was the largest band on the gel because it is the most abundant protein in egg white [54]. A fraction of protein aggregates that did not enter the electrophoresis gel (Figure 3, lanes 3 and 7) was presumably formed via disulfide bonds, as they disappeared under reducing conditions (Figure 3, lanes 4 and 8), as already mentioned by Alavi [56]. It can be noticed that the bands of ovotransferrin and ovalbumin appeared with a higher molecular weight under reducing conditions (Figure 3, lanes 3 vs. 4 at pH 7.5, lanes 7 vs. 8 at pH 9.0), probably due to the rupture of their internal disulfide bonds that expands their structure and thus increases their apparent molecular weight [57,58].

The polypeptide profile of PPI-EW mixtures at a weight ratio of 50/50 at pH 7.5 (lanes 1–4) and 9.0 (lanes 5–8) under reducing and non-reducing conditions with and without centrifugation is shown in Figure 4. Major protein components of PPI and EW were found under non-reducing conditions, such as vicilin (around 50 kDa, 20–37 kDa, 19 kDa), convicilin, legumin (Lαβ), ovalbumin, ovotransferrin, and lysozyme (Figure 4, lanes 1 and 8). When comparing the bands under reducing conditions between Figure 4 (lanes 2 and 7) and Figure 3 (lanes 2 and 4 and lanes 6 and 8 for PPI and EW, respectively), no band disappeared. However, regarding supernatant samples after centrifugation, the bands of vicilin at ~19 kDa, lipoxygenase and lysozyme (Figure 4, lanes 3 and 4), faded at pH 7.5 (Figure 4, lane 3) but not at pH 9 (Figure 4, lane 5) compared to initial mixture samples (Figure 4, lanes 1 and 2). This would suggest the contribution of these proteins in the insoluble part of the mixture in agreement with the observed loss of protein solubility around this pH (Figure 2, Table 1). In particular, some pea protein polypeptides could interact with lysozyme, which is positively charged around neutral pH, to form insoluble complexes [46]. Interestingly, this did not occur markedly at pH 9.0 (Figure 4, lane 5–8) when lysozyme was less positively charged at a closer pH to its pI (10.7).

### 3.3. Thermal Properties of the Mixtures

The thermal properties of proteins directly reflect their native status and can be evaluated by calorimetry. Figure 5A,B show typical DSC thermograms for pea proteins, egg white proteins, and PPI-EW mixtures at different weight ratios at pH 7.5 and 9.0, respectively. PPI thermal curves showed the two characteristic denaturation endothermic peaks for 7S and 11S globulins [59]. The first peak at around 71 °C corresponded to the denaturation of the lower molecular weight fraction (7S), and the second one at around 84 °C was related to the higher molecular weight fraction (11S). The respective areas of the two peaks were calculated by curve deconvolution and the area assigned to 7S (~58.8%) was much higher compared to the area assigned to 11S (~41.2%), confirming that 7S was the major fraction of globulins in our PPI sample. These results are consistent with data previously obtained by other authors on vicilin and legumin denaturation. For instance, O’Kane [60] reported that the denatured temperature of purified vicilin from pea protein was around 69.9–71.8 °C. O’Kane [60] illustrated that 11S legumin had a denaturation temperature of around 87 °C at a 0.5 °C/min heating rate. Figure 5B shows typical DSC thermograms for EW with four main peaks. In agreement with literature data [61,62], the peaks at ~63, ~69, ~76, and ~83 °C could be assigned to ovotransferrin, lysozyme, ovalbumin, and S-ovalbumin (the more heat-stable form of ovalbumin [26]), respectively.

The thermal denaturation temperature (Td) of PPI, EW, and PPI-EW mixtures at different weight ratios is compared at pH 7.5 and 9.0 in Table 2 and Table 3, respectively. In these tables, the peaks are assigned to the different proteins present in PPI, EW, and their mixtures. For PPI, Td3 and Td5 corresponded to vicilin and legumin, respectively. For EW, Td1, Td2, Td3, and Td4 corresponded to ovotransferrin, lysozyme, ovalbumin, and S-ovalbumin, respectively. However, at pH 7.5, the peak of lysozyme was overlayed by ovotransferrin one (Figure 5A), maybe due to co-aggregation and heteroprotein formation between ovotransferrin and lysozyme as suggested by Wei [28] and Iwashita [27]. Many studies performed on liquid egg white around neutral pH indeed mention two main denaturation peaks around 65 and 80 °C attributed to ovotransferrin and ovalbumin, respectively [63,64,65]. For the PPI-EW mixtures, Td3 resulted from the superimposition of ovalbumin and vicilin peaks, which could also overlay the peak of lysozyme at pH 9.0. Td4 and Td5 corresponded to S-ovalbumin and legumin denaturation, respectively.

At pH 7.5 (Table 2), no significant difference in the Td of ovotransferrin between pure EW and PPI-EW mixtures occurred. In the case of the EW sample, the first peak showed a shoulder (Figure 5A) probably corresponding to lysozyme. With the addition of PPI, the peak corresponding to lysozyme appeared distinctly. This may be due to either a slight shift in the ovotransferrin signal toward lower temperatures, thus resulting in a better separation of ovotransferrin and lysozyme signals, or to an increase in the lysozyme denaturation temperature due to its stabilization through interactions with PPI proteins as evidenced in our recent study [46]. As discussed before for solubility and SDS-PAGE results, pea proteins could form electrostatic interactions with lysozyme since they are strongly oppositely charged at pH 7.5. Moreover, the Td value of legumin (Td5) decreased by approximately 2 °C in PPI-EW mixtures compared to the pure PPI suspension. This means that legumin proteins were more sensitive to temperature in mixtures. Conformational changes toward a more unfolded state of legumin molecules could thus be hypothesized in the presence of egg white proteins. A modification of the hydration environment of molecules influenced by the new composition in the mixture and/or interactions with egg white proteins could explain the partial unfolding of legumin to a less stable form. As vicilin and ovalbumin had close denaturation temperatures, only one Td value (Td3 ≈ 76 °C) was recorded for the PPI-EW mixtures at pH 7.5. This value was not significantly affected by the PPI-EW weight ratio and corresponded to the mean of ovalbumin and vicilin Td values in pure EW and PPI systems, respectively. The Td of S-ovalbumin decreased slightly but not significantly for the mixtures compared to the pure EW sample.

At pH 9.0 (Table 3), the Td value of ovotransferrin (Td1) decreased significantly from ~63 to ~59 °C with the increase in PPI content in admixture, indicating that ovotransferrin was more sensitive to heat denaturation in the presence of PPI at pH 9.0 than at pH 7.5. This could be explained by the partial unfolding of ovotransferrin in the presence of PPI or a decrease in electrostatic interactions with lysozyme due to the competition with some pea proteins for complexation as suggested by the increase in the Td value of legumin in the mixture at this pH (Td5). No peak corresponding to lysozyme (Td2) was detected for PPI-EW mixtures at pH 9.0, although it was observed for pure EW samples at this pH. We assumed that the peak of lysozyme may be overlapped by the larger peak of ovalbumin and vicilin (Td3). In the presence of PPI, the total peak shifted just slightly to a lower Td value (approximately 1 °C lower). As in pure systems, Td values for 7S proteins (~71 °C) and lysozyme (69.5 °C) were significantly lower compared to the ovalbumin one (76.4 °C), where the superimposition of the three denaturation temperatures could explain the resultant average lower Td. It could also not be excluded that the 7S peak and/or lysozyme peak increased to Td values > 71 °C closer to the Td of ovalbumin in the mixture due to a cooperative denaturation effect or thermo-protective effect of the respective proteins. For instance, Mession [66] reported that, when mixing casein with pea vicilin-enriched fractions, the Td of the latter proteins increased by approximately 4 °C. In addition, Zheng [67] found that the Td temperature of the mixture of lysozyme and β-conglycinin was higher than lysozyme on its own, which indicates that the thermal stability of lysozyme was improved via the partial unfolding of β-conglycinin due to complexation. Finally, it is worth noting that the Td of S-ovalbumin showed no significant difference for all the samples at both pHs, which confirmed the high thermal stability of this protein.

Table 4 shows the specific denaturation enthalpy (∆H) values for suspensions of PPI and EW and their mixtures at different weight ratios at pH 7.5 and 9.0. The ∆H values were calculated according to Equation (3). It was found that the ∆H value of pure protein suspensions was influenced by pH. The ∆H value of the EW sample showed a slight but significant increase from pH 7.5 to 9.0 (from 22.3 to 23.8 J/g) while it presented a three-fold decrease for the PPI sample (from 10.8 to 3.6 J/g), respectively. The denaturation enthalpy (∆H) of the egg white agreed with previous data of 20.6 J/g at pH 7.0 obtained by Ferreira [62] and 21.0 J/g at pH 8.58 obtained by Rossi [68]. ∆H values of PPI were in the same order of magnitude as those obtained by Sun [69], i.e., around 8.3 J/g of salt-extracted pea proteins, or Mession [70], i.e., 11.4 J/g of acid-precipitated pea proteins at pH 7.5. The ∆H value of the PPI sample at pH 7.5 in the present study is an indication that the globulin fractions produced were low-denatured. However, the ∆H decrease at pH 9.0 compared to pH 7.5 indicates that increasing intramolecular net charges and repulsive interactions at this more alkaline pH, far from the pI of pea proteins, caused the partial unfolding of protein as similarly demonstrated by Meng [71] on red bean globulins.
(3)$$∆H:{∆H}_{calculated}={∆H}_{ew}\times ratio+{∆H}_{ppi}\times ratio$$

Regarding the PPI-EW mixtures at different weight ratios, the ∆H value was calculated by Equation (3) to know if the measured values of ∆H resulted from the additive denaturation of PPI and EW proteins considering their relative content in the mixtures. Mixtures at pH 7.5 presented measured ∆H values significantly lower than calculated ones with differences comprised between 0.8 and 2.5 J/g. These differences could reflect the loss of the solubility of some proteins as revealed for the 50/50 mixture in previous Section 3.1 and Section 3.2 at this pH as the precipitated protein part was less prone to contributing to the total enthalpy.

The PPI-EW mixture at 25/75 and 75/25 ratios at pH 9.0 showed calculated ∆H values similar to measured ones, which could probably indicate that the interactions that structure the different proteins were not significantly modified in the mixture in these conditions. However, the 50/50 ratio at pH 9.0 showed a slightly lower ∆H value than the calculated one, which probably originated due to a small loss of solubility compared to an ideal additive system (Figure 2, Table 1).

### 3.4. Gelation Temperatures

Temperature sweeps were performed by small amplitude rheology to understand the sol–gel transition behavior of the different protein suspensions upon heating. The gelling temperatures are reported in Table 5. For EW and the PPI-EW mixtures containing at least 50% EW, two transition temperatures were measured regardless of the pH. For pure EW at pH 7.5 and 9.0, the two gelling temperatures at ~60 and ~75 °C could be attributed preferentially to the denaturation of ovotransferrin and ovalbumin, respectively, as reported by Barhut [61] and Ferreira [62]. These temperatures were indeed close to the Td values identified for these proteins from DSC thermograms (Table 2 and Table 3). For the mixtures containing at least 50% EW, considering that pure PPI suspensions did not show any transition temperature around 60 °C regardless of the pH, the presence of a first gelling point can also be associated mainly to ovotransferrin. Compared to whey–pea proteins mixtures already studied [72,73], this early gel point could not be observed as the sol–gel transition temperature of whey proteins seems rather similar to ovalbumin behavior [74].

The first transition temperature was not affected by the EW/PPI ratio at pH 7.5 whereas it increased by ~3 °C with the increased PPI content in the mixture at pH 9.0. At this latter pH, the gelation point assigned to ovotransferrin was therefore delayed even though the DSC results reported in Section 3.3 show a slight decrease in the denaturation temperature of this protein in admixture with PPI. The presence of the pea globulins carrying highly negative charges at this pH, far from their pI, could be prone to prevent ovotransferrin molecules/particles association until more advanced denaturation (or aggregation) is achieved at slightly higher temperatures. Similar behavior was observed by Watanabe [75] with dry-heated ovalbumin inhibiting ovotransferrin heat aggregation and coagulation.

In addition, the PPI-EW mixtures at the 75/25 weight ratio at pH 7.5 and 9.0 did not show any sharp G′ rise around 60 °C, meaning that no early sol–gel transition can be associated with ovotransferrin in this mixture. It could be hypothesized that, even if the thermal denaturation of ovotransferrin occurred around 60 °C, the resulting unfolded/aggregated proteins were not numerous enough to result in a three-dimensional network and/or their association was sterically hindered by the presence of the pea globulins in the mixture.

The second transition temperature was observed in all cases except for the pure PPI sample at pH 9.0, which did not reach gelation during the heating ramp. In this last case, the formation of a three-dimensional network of unfolded/aggregated pea globulins was hindered by high repulsive forces within protein particles because the negative charges dominated at this pH far from the pI of globulins (pI = 4.5–4.8). For all mixtures, the second sol–gel transition around 75 °C was ascribed to ovalbumin and 7S globulins, which denatured in the same range of temperature as previously evidenced by DSC analysis. No sol–gel transition was specifically assigned to legumin, which presented a maximum denaturation temperature of around 85 °C as observed in Section 3.3. This range of temperature rather corresponded to a slowing threshold of G′ toward maximum stable values similarly to what was observed in Figure 1 for the pure EW system.

In order to consider the overall contribution of all proteins in the systems upon thermal treatment, we measured the final values of G′ and G″ at the end of heating (Table 6). In a first approach, these values could be representative of the level of association of the proteins undergoing denaturation up to 95 °C. Moreover, they could indicate that mainly the contribution of hydrophobic interactions and covalent SS bond formation could be considered in thermal protein aggregation upon temperature sweep, as electrostatic interactions such as hydrogen bonds considerably weakened in this temperature range. The works of Wang [76] on wheat gluten gel formation and Chronakis [77] on the formation of Spirulina protein thermal gels indeed highlight that disulfide bonds and hydrophobic interactions dominated during the heating process while hydrogen bonds and electrostatic interactions did not significantly contribute to gel formation but may reinforce the network rigidity of the protein network on cooling.

Except for the pure PPI sample at pH 9.0 that did not gel, all the protein systems presented a final G′ >> G″ by a factor of approximately 10. The G′ and G″ values decreased significantly and gradually when the proportion of EW decreased in the protein suspensions. Considering that, for the pure PPI sample at pH 7.5, the viscoelastic parameters were very low and no gel was formed at pH 9, it could be deduced that the sol–gel transition upon heating in the EW-containing systems mostly reflects the contribution of EW proteins. Native EW proteins were able to gel at alkaline pH as reported in other studies [78,79]. The decrease in G′ values when adding PPI was explained by a lower concentration of EW in the system and a possible steric hindering effect caused by pea globulin unfolded/aggregated molecules formed all along the temperature sweep. As already indicated, the repulsive force between pea proteins at the pHs used was not favorable to the aggregation of globulin molecules neither in the pure PPI sample nor in PPI-EW mixtures. Regarding the pH effect, the G′ and G″ values were significantly higher and lower at pH 9.0 (vs. pH 7.5) for pure EW samples and PPI-EW mixtures, respectively (Table 6). It has already been reported that EW gels present a higher gel strength when prepared at a pH around 9.0 compared to a lower pH [78,79]. The opposite behavior between pH 9.0 and pH 7.5 observed in the case of PPI-EW mixtures confirmed the hindering effect of highly repulsive pea protein particles on the EW protein association. This effect was already found favorable for delaying the sol–gel transition of ovotransferrin at pH 9.0 as supposed when considering the first gel point in the system. This result could be considered positive in order to apply a higher treatment (e.g., pasteurization) temperature at an elevated pH during ingredient processing.

## 4. Conclusions

The thermal parameters (Td of each protein and total ∆H) of the composite protein systems were found to be slightly or not different compared to the pure protein suspensions of EW and PPI. The slight differences observed for ∆H could be explained by a limited loss (<10%) of protein solubility in the mixtures. However, the slight shifts in the Td value observed for some proteins in the mixed system could be explained by interactions between some proteins (mainly lysozyme with ovotransferrin and/or pea protein) acting positively or negatively on the thermal stability of the proteins depending on the pH. The pH indeed played a significant role. Indeed, the partial unfolding and high charge of pea protein at alkaline pH (pH 7.5 and 9.0) could affect the protein association upon heating. The heat-induced gelation behavior of the PPI-EW mixtures seemed governed by the EW proteins. However, the presence of pea proteins that underwent denaturation but insufficient aggregation due to high repulsive forces at an elevated pH was supposed to delay the self-association of EW proteins forming the nascent gel network. For instance, the primary sol–gel transition of ovotransferrin around 60 °C was slightly delayed by ~3 °C at pH 9.0. Thus, the use of pH 9.0 should be considered to optimize the heat treatment of the PPI-EW mixtures for the production of a composite ingredient. Further investigations into the thermal coagulation and foaming properties of these protein mixtures are also expected for adequate applications in food.

## Figures and Tables

**Figure 1 foods-12-02528-f001:**
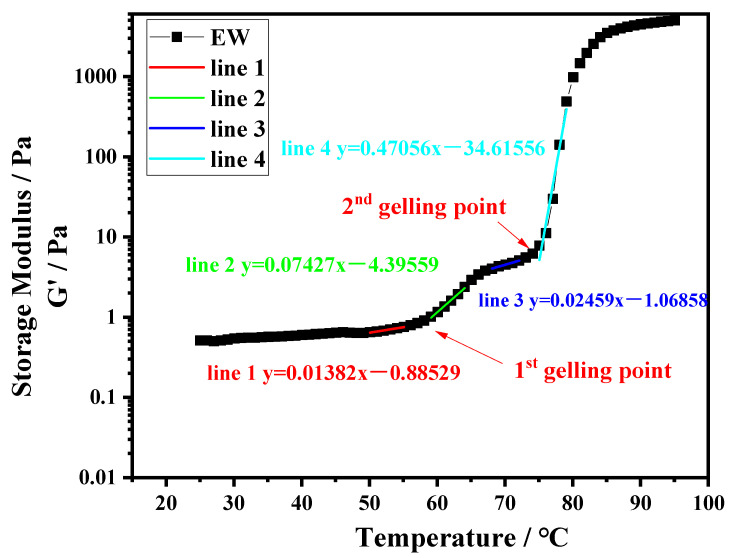
Temperature sweep for EW at pH 9.0.

**Figure 2 foods-12-02528-f002:**
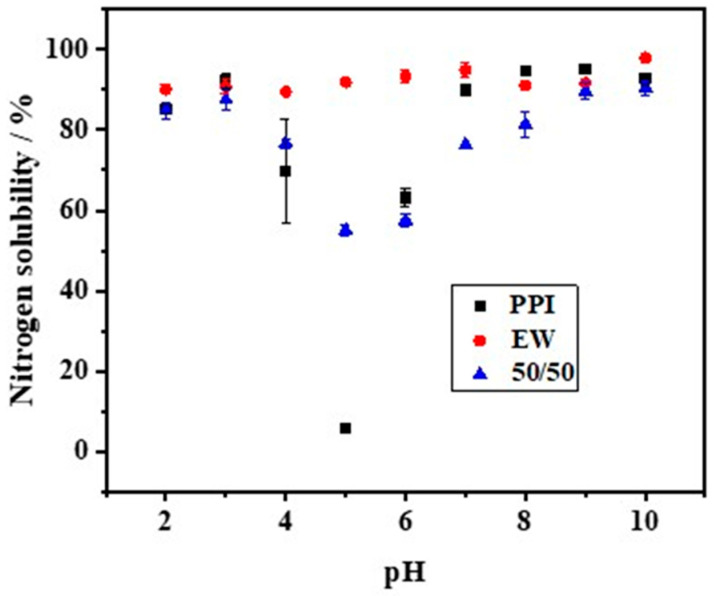
Nitrogen solubility of PPI, EW, and PPI-EW mixture at the 50/50 weight ratio in 0.1 M NaCl.

**Figure 3 foods-12-02528-f003:**
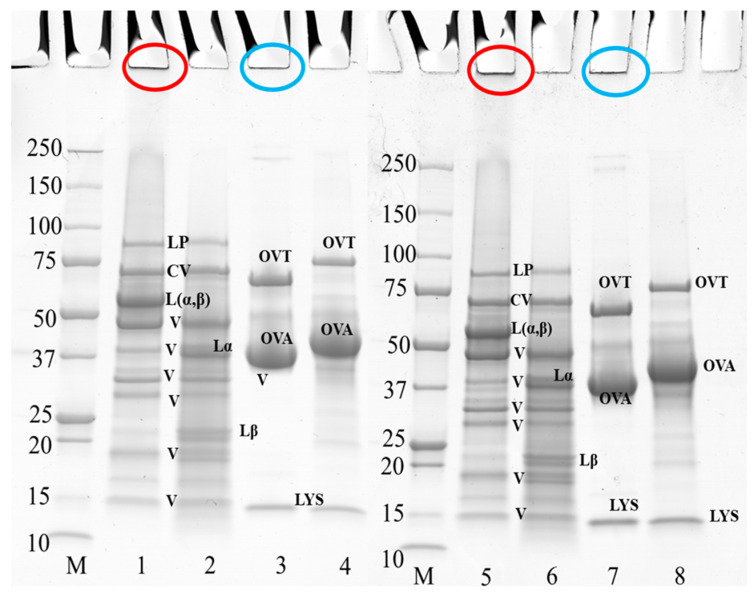
SDS-PAGE profile of 10 wt% PPI suspension and EW when prepared at pH 7.5 and 9.0. The samples on lanes 2, 4, 6, and 8 were treated under reducing conditions with SDS + DTT reagents. Circles in red and blue at the top of the bands indicate the aggregates of PPI and egg white, respectively. Lanes 1–4 at pH 7.5, lanes 5–8 at pH 9.0; lane M: molecular weight (Mw) markers; lanes 1–2, and 5–6: PPI; lanes 3–4, and 7–8: EW. LP, lipoxygenase; L (α, β), legumin; CV, convicilin; Lα, legumin acid subunit; Lβ, legumin basic subunit; V, vicilin; OVA, ovalbumin; OTA, ovotransferrin; LYS, lysozyme.

**Figure 4 foods-12-02528-f004:**
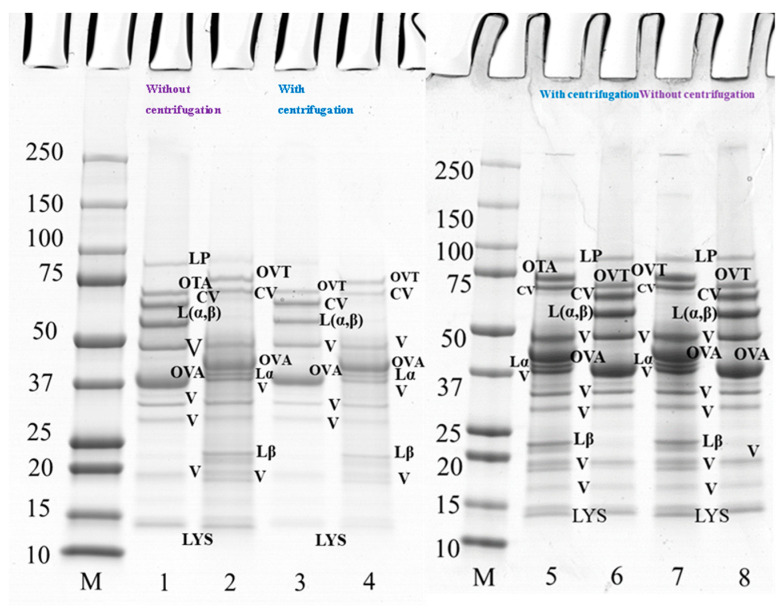
SDS-PAGE profile of 10 wt% protein suspension of PPI-EW mixture at 50/50 weight ratio with and without centrifugation prepared at pH 7.5 and 9.0. The samples on lanes 2, 4, 5, and 7 were treated under reducing conditions with SDS + DTT reagents. Lanes 1–4 at pH 7.5, lanes 5–8 at pH 9.0. Lane M: molecular weight (Mw) markers; lanes 1–2, 7–8: PPI-EW at the weight ratio 50/50 non-centrifugation; lanes 3–4, 5–6: PPI-EW at the weight ratio 50/50 with centrifugation. LP, lipoxygenase; L(α, β), legumin; CV, convicilin; Lα, legumin acid subunit; Lβ, legume basic subunit; V, vicilin; OVA, ovalbumin; OTA, ovotransferrin; LYS, lysozyme.

**Figure 5 foods-12-02528-f005:**
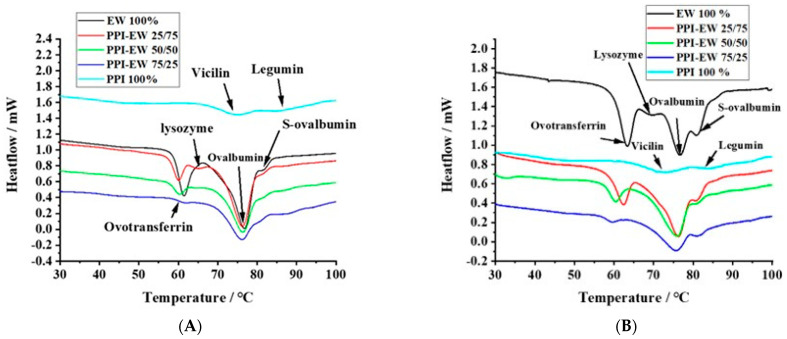
Typical DSC thermograms for PPI, EW, and PPI-EW at different weight ratios at pH 7.5 (**A**) and 9 (**B**), respectively.

**Table 1 foods-12-02528-t001:** Nitrogen solubility (NS) values of the PPI-EW mixture at the 50/50 weight ratio at different pHs. Calculated NS values were obtained from experimental NS values of individual PPI and EW suspensions.

pH	NS (%)	Calculated NS (%)
2	84.8 ± 1.2 a	87.6 ± 0.5 a
3	87.5 ± 1.4 a	91.5 ± 1.3 a
4	76.6 ± 0.7 a	78.3 ± 1.6 a
5	55.0 ± 0.8 a	48.9 ± 0.2 b
6	57.5 ± 0.8 a	78.6 ± 1.3 b
7	76.2 ± 0.1 a	92.5 ± 0.7 b
8	81.3 ± 1.8 a	92.9 ± 0.4 b
9	89.7 ± 1.2 a	93.4 ± 0.3 b
10	90.5 ± 1.0 a	95.3 ± 0.5 b

Means followed by a different lowercase letter for the same row are significantly different.

**Table 2 foods-12-02528-t002:** Thermal denaturation temperatures of EW, PPI, and PPI-EW mixtures at different weight ratios at pH 7.5.

Samples	Td 1 (°C)	Td 2 (°C)	Td 3 (°C)	Td 4 (°C)	Td 5 (°C)
EW 100%	61.1 ± 0.1 a	-	76.7 ± 0.1 a	83.5 ± 0.8 a	-
PPI-EW 25/75	60.1 ± 0.1 a	64.4 ± 0.6 a	76.0 ± 0.1 ab	81.6 ± 0.1 a	85.4 ± 0.3 a
PPI-EW 50/50	60.5 ± 0.4 a	64.5 ± 0.6 a	76.2 ± 0.1 ab	81.5 ± 0.1 a	85.6 ± 0.1 a
PPI-EW 75/25	61.4 ± 0.1 a	-	76.0 ± 0.1 ab	81.3 ± 0.2 a	85.0 ± 0.7 a
PPI 100%	-	-	75.8 ± 0.4 b	-	87.4 ± 0.5 b

All data are given as mean ± SD of triplicate measurements. Means in a column bearing the same letter are not significantly different. Td 1: ovotransferrin or ovotransferrin and lysozyme; Td 2: lysozyme; Td 3: ovalbumin (EW), vicilin (PPI) or ovalbumin and vicilin (mixture case), Td 4: s-ovalbumin; Td 5: legumin.

**Table 3 foods-12-02528-t003:** Thermal denaturation temperatures (Td) of EW, PPI, and their mixtures (PPI-EW) at different weight ratios at pH 9.0.

Samples	Td 1 (°C)	Td 2 (°C)	Td 3 (°C)	Td 4 (°C)	Td 5 (°C)
EW 100%	63.2 ± 0.1 a	69.5 ± 0.1	76.4 ± 0.1 a	83.1 ± 0.7 a	-
PPI-EW 25/75	62.1 ± 0.1 b	-	76.1 ± 0.1 ab	81.3 ± 0.1 a	86.3 ± 0.7 a
PPI-EW 50/50	60.3 ± 0.1 c	-	75.3 ± 0.2 b	81.4 ± 0.1 a	86.1 ± 0.1 a
PPI-EW 75/25	59.5 ± 0.1 d	-	75.5 ± 0.1 b	81.4 ± 0.1 a	86.9 ± 0.2 a
PPI 100%	-	-	71.3 ± 0.4 c	-	84.5 ± 0.2 b

All data are given as mean ± SD of triplicate measurements. Means in a column bearing the same letter are not significantly different. Td 1: ovotransferrin or ovotransferrin and lysozyme; Td 2: lysozyme; Td 3: ovalbumin (EW), vicilin (PPI), or ovalbumin, vicilin, and lysozyme (mixture case), Td 4: s-ovalbumin; Td 5: legumin.

**Table 4 foods-12-02528-t004:** Denaturation enthalpy (∆H) of EW, PPI, and their mixtures (PPI-EW) at different weight ratios at pH 7.5 and 9.0.

	pH 7.5		pH 9	
Enthalpy (J/g Protein)	∆H	Calculated ∆H	∆H	Calculated ∆H
EW 100%	22.3 ± 0.5 *	-	23.8 ± 0.2 **	-
PPI-EW 25/75	18.6 ± 0.2 a	19.4 ± 0.4 b	18.3 ± 0.3 a	18.7 ± 0.1 a
PPI-EW 50/50	14.1 ± 0.2 a	16.6 ± 0.2 b	12.4 ± 0.1 a	13.7 ± 0.1 b
PPI-EW 75/25	12.6 ± 0.1 a	13.7 ± 0.1 b	8.6 ± 0.1 a	8.6 ± 0.2 a
PPI 100%	10.8 ± 0.1 *	-	3.6 ± 0.2 **	-

All data are given as mean ± SD of triplicate measurements. Means in a row bearing the same letter are not significantly different at the same pH. Means followed by different numbers of * for the same row are significantly different.

**Table 5 foods-12-02528-t005:** Gelling point temperature of PPI, EW suspensions, and their mixtures at 10 wt% protein at pH 7.5 and 9.0.

Samples	pH 7.5		pH 9	
	1st Gelling Point/°C	2nd Gelling Point/°C	1st Gelling Point/°C	2nd Gelling Point/°C
EW 100%	59.3 ± 0.2 a	75.4 ± 0.3 a	58.8 ± 0.3 a	75.2 ± 0.4 ab
PPI-EW 25/75	59.3 ± 0.2 a	75.13 ± 0.3 a	60.7 ± 0.3 c	75.5 ± 0.4 a
PPI-EW 50/50	59.9 ± 0.2 a	75.07 ± 0.3 a	61.6 ± 0.2 c	74.0 ± 0.3 b
PPI-EW 75/25	none	73.2 ± 0.3 b	none	75.6 ± 0.3 a
PPI 100%	none	75.6 ± 0.3 a	no gel	

All data are given as mean ± SD of triplicate measurements. Means in a column bearing the same letter are not significantly different.

**Table 6 foods-12-02528-t006:** Final G′ and G″ values at 95 °C of PPI, EW, and PPI-EW mixtures at 10 wt% protein at different weight ratios at pH 7.5 and 9.0.

Samples	G′ (Pa)		G″ (Pa)	
	pH 7.5	pH 9	pH 7.5	pH 9
EW 100%	4865 ± 156 a	5496 ± 131 a	425 ± 21 a	369 ± 6 a
PPI-EW 25/75	2552 ± 149 b	2077 ± 117 b	210 ± 17 b	167 ± 8 b
PPI-EW 50/50	1189 ± 100 c	953 ± 65 c	137 ± 26 bc	83 ± 1 c
PPI-EW 75/25	742 ± 181 c	106 ± 6 d	94 ± 29 c	17 ± 1 d
PPI 100%	30 ± 26 d	no gel	2 ± 1 d	no gel

All data are given as mean ± SD of triplicate measurements. Means in a column bearing the same letter are not significantly different.

## Data Availability

Data is contained within the article.

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
