# Peer review of "Thermal Behavior of Pea and Egg White Protein Mixturesâ€"

_foods, 2023, doi:10.3390/foods12132528_

Round 1
Reviewer 1 Report
The manuscript examines the thermal and rheological properties of the mixtures of egg white protein with pea proteins. It was well organized and has a logic relation through the paper. The English structure is OK for the journal. There are a few concerns needs to be addressed as follows:
Line 19: what is admixture?
Line 112: please verify the specification of centrifuge.
Line 117: please provide more detail about the UF processing.
Line 121: Provide the specification of Freeze-dryer.
Line 138: eight ratios of ….what is the 8 ratios?
Section 3.4. It is strongly recommend compare the results of the gelling point of the protein mixture with whey protein.
The English structure is OK for the journal.
Author Response
We would like to thank all the reviewers for their comments. All the modifications appear in red in the revised version.
# Reviewer 1
« The manuscript examines the thermal and rheological properties of the mixtures of egg white protein with pea proteins. It was well organized and has a logic relation through the paper. The English structure is OK for the journal. There are a few concerns needs to be addressed as follows:
Line 19: what is admixture? »
Admixture means mixture or blend ; « in admixture » is an expression indicating that the compounds are associated in the same mixture.
« Line 112: please verify the specification of centrifuge. »
The specification of the centrifuge was added lines 114-115.
« Line 117: please provide more detail about the UF processing. »
The information was added lines 120-125.
« Line 121: Provide the specification of Freeze-dryer. »
The information regarding freeze-drying procedure was added lines 125-129.
« Line 138: eight ratios of ….what is the 8 ratios? »
We checked; the terms « weight ratio » was correct, and not « eight ratio ».
« Section 3.4. It is strongly recommend compare the results of the gelling point of the protein mixture with whey protein. «
A sentence was added lines 465-467 in the discussion to compare the sol/gel transition temperature of whey-pea protein mixtures considered in some previous works.
Reviewer 2 Report
The influences of pH on the structure of main components of PPI and EW and their further influences on the gelling properties are suggested to be investigated.
The manuscript is scientific.
The method part of the manuscript is appropriate.
The discussion of the results is need to be improved.
It interesting is average. The application of this work is suggested to be introduce more.
It expanded the existing knowledge.
Compared with other published material, it add the differential scanning calorimetry information to the subject area.
The paper is well written.
The text is clear and easy to read.
The conclusions are consistent with the evidence and arguments presented.
They addressed the main question posed.
Author Response
We would like to thank all the reviewers for their comments. All the modifications appear in red in the revised version.
« The influences of pH on the structure of main components of PPI and EW and their further influences on the gelling properties are suggested to be investigated. »
Thanks for the suggestion; we are working on a new paper specifically considering the gelling properties of the system: texture analysis, dynamic rheology, gel microstructure by confocal microscopy, molecular interactions driving the protein network formation…
“The manuscript is scientific.
The method part of the manuscript is appropriate.
The discussion of the results is need to be improved.”
It is difficult to understand what is precisely required by reviewer 2. As recommender by the reviewer 1 we add a comparison with whey/pea proteins mixtures in the discussion section.
“It interesting is average. The application of this work is suggested to be introduce more.”
In modified lines 41-42 & 84-85, we insisted again that the perspective of the paper is the development of a mixed EW-PPI ingredient undergoing similar treatments as for commercial egg products as explained lines 78-84.
“It expanded the existing knowledge.
Compared with other published material, it add the differential scanning calorimetry information to the subject area.
The paper is well written.
The text is clear and easy to read.
The conclusions are consistent with the evidence and arguments presented.
They addressed the main question posed.”
Reviewer 3 Report
The manuscript presented for publication investigates the protein modifications in hybrid system of pea and egg white. The authors looked a different ratio of pea and egg white proteins and their behaviours under thermal treatment. The article is nicely organized and written. Material and Methods is very clear and analysis are complete. Results and discussion are appropriate and of good quality. I have no specific comment on the manuscript, except for Table 1, where the first row needs to be edited (replace title 1, title 2 and title 3 with correct titles).
Author Response
We would like to thank all the reviewers for their comments. All the modifications appear in red in the revised version.
« The manuscript presented for publication investigates the protein modifications in hybrid system of pea and egg white. The authors looked a different ratio of pea and egg white proteins and their behaviours under thermal treatment. The article is nicely organized and written. Material and Methods is very clear and analysis are complete. Results and discussion are appropriate and of good quality. I have no specific comment on the manuscript, except for Table 1, where the first row needs to be edited (replace title 1, title 2 and title 3 with correct titles). »
Thank you for the relevant comment. We edited Table 1 with the correct column titles.